

# The emerging roles of circHECTD1 in human diseases and the specific underlying regulatory mechanisms

Yiran Yuan[1,*], Xiaomin Zhang[1,*], Xiaoxiao Wang[1], Lei Zhang[2,3] and Jiefeng He[1,2]

[1] Third Hospital of Shanxi Medical University, Shanxi Bethune Hospital, Shanxi Academy of Medical Sciences, Tongji Shanxi Hospital, Taiyuan, Shanxi, China

[2] Department of Hepatobiliary Surgery, Shanxi Bethune Hospital, Shanxi Academy of Medical Sciences, Tongji Shanxi Hospital, Third Hospital of Shanxi Medical University, Taiyuan, Shanxi, China

[3] Hepatic Surgery Center, Institute of Hepato-Pancreato-Biliary Surgery, Tongji Hospital, Tongji Medical College, Huazhong University of Science and Technology, Wuhan, Hubei, China

[*] These authors contributed equally to this work.

Corresponding authors
Lei Zhang, zhangl7803@126.com
Jiefeng He, jfhe@sxmu.edu.cn

## ABSTRACT

Circular RNAs (circRNAs) are a class of single-stranded closed-loop RNAs that have become a popular research subject in biology. Compared to linear RNAs, they are more stable, more conserved, and more widely distributed, and they play crucial biological functions in many diseases. CircHECTD1, a newly identified member of the circRNA family, is widely distributed in humans. Recent studies have shown that circHECTD1 is abnormally expressed in various human diseases, including glioma, hepatocellular carcinoma, gastric cancer, acute ischaemic stroke, silicosis, acute lung injury, ulcerative colitis, atherosclerosis, and hypertrophic scarring. In malignant tumours, circHECTD1 is thought to be an oncogene that promotes malignant tumour behaviours and influences tumour prognosis. In nontumour diseases, it plays a dual role, promoting disease in silicosis, stroke, and other diseases, while alleviating the disease process in ulcerative colitis, acute lung injury, and atherosclerosis. This article provides a review of the regulatory roles and mechanisms of action of circHECTD1 in different diseases. We also discuss and prospectively evaluate the clinical potential of circHECTD1 as a diagnostic biomarker and a therapeutic target for related diseases, providing new insights for developing new therapeutic strategies.

## INTRODUCTION

The investigation of disease causation has been ongoing since ancient times, with attention shifting from macroscopic environmental factors to microscopic molecular mechanisms (*Genuis, 2012*). Since the 1970s, research has focused mainly on mRNAs that encode proteins for the causes of diseases and potential therapeutics. However, with advancements in technology, researchers have begun to explore other ncRNAs, a broader and more diverse class of RNAs that do not encode proteins (*Matsui & Corey, 2017*). Less than 2% of the human genome encodes proteins, while over 98% does not (*Poller et al., 2018*). NcRNAs are a general term for a class of RNAs consisting of microRNAs (miRNAs),

long noncoding RNAs (lncRNAs), intronic RNAs, and repetitive RNAs (*Shi et al., 2021*). NcRNAs may not encode proteins, but they play significant roles in cellular physiology and pathological regulation through several pathways, including gene transcription and translation regulation, modulation of mRNA targets, influence on cell proliferation and differentiation, and cytokine secretion (*Lodde et al., 2022*). Additionally, abnormal expression of ncRNAs is consistently associated with disease and plays critical roles in several human illnesses, such as cancers, cardiovascular diseases, inflammation, and neurodegenerative diseases (*Bhatti et al., 2021*; *Esteller, 2011*).

CircRNAs are a special class of lncRNAs with closed-loop structures. Due to their unique structures, these RNAs are more stable and have longer biological half-lives than linear RNAs. This enables them to perform functions that cannot be performed due to the short half-life of linear RNAs (*Jeck & Sharpless, 2014*; *Zhou et al., 2020*). CircRNAs are widely distributed in the human body and play various biological regulatory roles, such as acting as miRNA sponges to suppress miRNAs (*Caba et al., 2021*). As a result, circRNAs are often considered oncogenes that promote malignant behaviours such as proliferation, migration, invasion, and drug resistance in tumour cells (*Li et al., 2022b*). Among emerging circRNAs, circHECTD1 has gained significant attention. Numerous studies have shown that circRNAs play a crucial role in the regulation of tumour development (*Dong et al., 2023*), and that they also exert important regulatory functions in nontumour contexts (*Kristensen et al., 2019*). Among them, circHECTD1 has been mentioned multiple times in research on both tumour and nontumour diseases (*Han et al., 2018*; *Jiang et al., 2020*), indicating its enormous potential in disease research. A deeper understanding of the function, structure, and regulatory mechanisms of circHECTD1 in diseases can provide a new perspective for RNA-level research and offer new ideas for the development of drugs based on circRNAs. The following sections provide detailed descriptions of the structure and functions of circRNAs and the regulatory mechanisms of circHECTD1 in various human diseases.

## Intended audience and need for this review

CircRNAs are multifunctional noncoding RNAs with covalent ring structures, and have emerged as a hot research topic in recent years. Among the newly discovered circRNAs, circHECTD1 has garnered increasing attention, yet current studies on its expression and function are not comprehensive. Therefore, this review briefly summarizes the expression and function of circHECTD1 in various tumour and nontumour diseases, focusing on its role and regulatory mechanisms in disease development. Additionally, we highlight the potential of circHECTD1 in disease diagnosis and prognosis, as well as the challenges faced in human disease research. The broad and comprehensive expression of circHECTD1 in the human body is crucial for diagnosing and predicting various diseases. Further preclinical studies are needed to explore its potential as a biomarker for diagnosing, treating, and predicting various human diseases.

In summary, this article aims to provide researchers with an overview of the latest research progress on the role of circHECTD1 in various tumours and nontumour diseases while offering valuable insights for future research directions.

## Survey methodology

To ensure an inclusive and unbiased analysis of the literature and to achieve the review objectives, we conducted a thorough search of the PubMed, Web of Science, and China National Knowledge Infrastructure databases. The search terms included: circHECTD1, circular RNA HECTD1, circRNA, circular RNA, noncoding RNA, circular intronic RNA, exon–intronic circRNA, circRNA-protein interaction, function, mechanism, glioma, gastric cancer, hepatocellular carcinoma, ischaemic stroke, silicosis, acute lung injury, ischaemia-reperfusion injury, ulcerative colitis, hypertrophic scar, and drug resistance. In this article, based on published literature, keywords and their variants as well as related words were sorted, collated and then searched. Ultimately, a total of 147 relevant studies were comprehensively selected after excluding duplicate searches and those with little relevance.

# CIRCRNAS

## Origin and structure of circRNAs

CircRNAs are a special type of lncRNA with a single-stranded covalent closed-loop structure (*Zhou et al., 2020*). Although they were first discovered to exist in plant cells in 1976 (*Sanger et al., 1976*), the level of knowledge and technical conditions at that time led to them being considered transcriptional shear by products, and they did not attract much attention (*Li, Ma & Li, 2019a*). In recent years, due to advancements in RNA high-throughput sequencing technology and bioinformatics, circRNAs have been shown not to be "transcriptional waste", and more attention has been given to studying their origin, structure, and functions. The single-stranded covalent closed-loop structure of circRNAs is mainly formed by covalent bonding of the 3′ end of the middle exon and the adjacent 5′ end of the precursor mRNA (pre-mRNA) during reverse shear, instead of the canonical shearing of normal linear RNAs (*Kumar et al., 2017*). As a result, circRNAs do not possess a free 5′ terminal cap structure or a 3′ terminal poly A structure, which renders them more resistant to nucleic acid exonucleases than linear RNAs, resulting in greater stability, conserved properties, and a longer biological half-life (*Jeck & Sharpless, 2014*; *Lei et al., 2019*). CircRNAs can be broadly classified into four types based on the origin and sequence of circRNAs: (1) exonic circular RNAs (ecircRNAs), which are formed by exon circularization, consisting of single or multiple exons of a gene and accounting for the majority of circRNAs; (2) circular intron RNAs (ciRNAs); (3) exon–intron circular RNAs (EIciRNAs), which contain both exons and introns(Fig. 1); and (4) studies that have also reported the formation of circRNAs from genes encoding viral RNA, snRNA, tRNA, and other RNAs (*Li et al., 2022b*; *Rao et al., 2021*).

## Functions of circRNAs

Different circRNAs are localized differently within cells: ecircRNAs are mainly found in the cytoplasm and exosomes, whereas ciRNAs and EIciRNAs are mainly found in the nucleus (*He et al., 2021a*). This distribution pattern indicates that circRNAs can regulate gene expression at various levels, including epigenetics, gene transcription, posttranscriptional

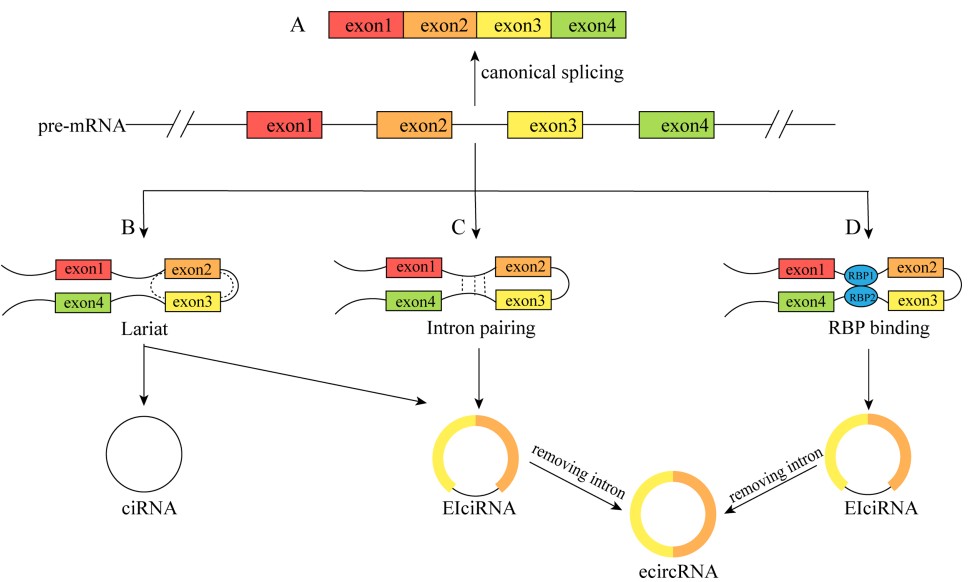

**Figure 1** **The formation process of the circRNAs.** (A) Canonical splicing to form mRNA; (B) cyclization driven by lariat; (C) intron pairing drives cyclization; (D) RBP was combined with cyclization to form a bridge. The pre-mRNA was spliced to produce an RBP binding RNA containing exons 2 and 3 and then generated ciRNA and EIciRNA. Furthermore, EIciRNA removed intron to form ecircRNA.

modifications, and protein translation. CircRNAs play roles in regulating intracellular pre-mRNA splicing and transcription by acting as miRNA sponges to regulate the expression of their target genes, encoding proteins, facilitating protein–protein interactions, and other cellular physiological processes (*Caba et al., 2021*; *Zhou et al., 2021*).

### circRNAs, as competing endogenous RNAs (ceRNAs) or miRNA molecular sponges, regulate the expression of miRNA-targeted genes

The miRNA sponge role of circRNAs is a popular area of research. miRNAs can regulate mRNA stability and translation by complementary binding to the 3′-untranslated regions (UTRs) of mRNAs, resulting in reduced target protein synthesis without affecting mRNA expression levels (*Pillai, 2005*). However, circRNAs have abundant miRNA-binding sites and can act as ceRNAs to competitively repress miRNAs or as molecular sponges to adsorb miRNAs and thus upregulate the protein expression of their target genes (*Panda, 2018*). circHIPK3 inhibits its complementary binding to IGF1 mRNA by sponging miR-379 and upregulates IFG1 protein expression thereby promoting the development of non-small cell lung cancer (NSCLC) (*Tian et al., 2017*). *Wang et al. (2019)* reported that circHIAT1 acts as a miR-3171 sponge to upregulate PTEN in hepatocellular carcinoma (HCC), thereby promoting HCC development. The role of miRNA sponges of circRNAs suggests that circRNAs can regulate the level of gene expression, which is not only related to the development of various diseases, but also facilitates further research on the molecular regulation of diseases and improves the understanding of the molecular regulatory network of diseases (Fig. 2A).

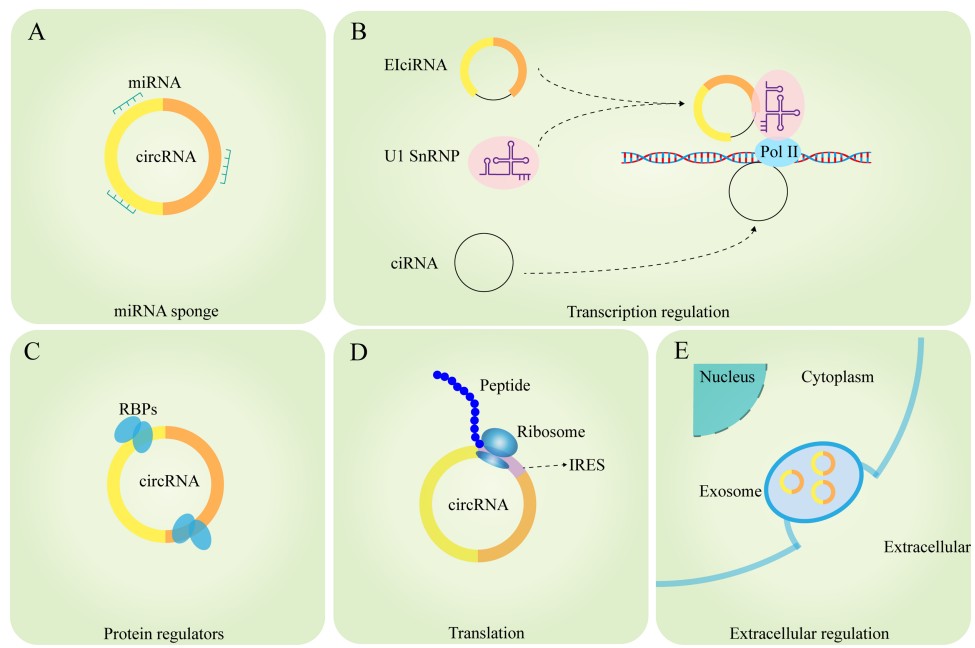

**Figure 2 Functions of circRNAs.** (A) circRNAs can serve as miRNA sponges to regulate the expression of target genes; (B) the ElciRNAs forms a complex with U1 snRNP, which then binds to Pol II in the promoter region to regulate the transcription level of genes; ciRNAs can directly accumulate in the parental transcribed region to promote Pol Il activity and regulate the expression of downstream genes; (C) circR-NAs can interact with RBPs to regulate the function of related proteins; (D) circRNAs can serve as templates for the translation of peptides or proteins; (E) circRNAs can exert extracellular regulatory functions through exosomes.

### circRNAs promote RNA polymerase II-mediated gene transcription

It has been shown that EIciRNAs are abundantly enriched in the nucleus, and EIciRNAs promote gene expression by forming the eiciRNA-U1 SnRNP complex through RNA–RNA interactions with U1 SnRNA, which then interacts with the RNA polymerase II transcriptional complex on the parental gene promoter (*Li et al., 2015*). *Zhang et al. (2013)* found that ciRNA-ANKRD52 could interact with RNA polymerase II to promote transcription, and knockdown of ciRNA-ANKRD52 would not affect the binding of ciRNA to RNA polymerase II or the expression of ANKRD52 but would reduce the rate of ANKRD52 gene transcription (Fig. 2B).

### circRNAs regulate protein function by interacting directly with proteins

Many circRNAs have been found to regulate protein functions through interactions with RNA-binding proteins (RBPs). For example, circ-Amotl1 can interact with phosphoinositide dependent kinase (PDK1) and serine/threonine kinase (AKT1), leading to the promotion of pAKT expression and nuclear ectopia in primary cardiomyocytes and protection against myocardial injury caused by anthracycline doxorubicin (*Zeng et al., 2017*). CircARSP91 can bind to UL16-binding protein 1 (ULBP1), promoting the cytotoxicity of natural killer cells against HCC. CircARSP91 can also bind to adenosine deaminase that acts on RNA (ADAR1), inhibiting HCC growth (*Ma et al., 2019*; *Shi et*

*al., 2017*). CircRNA-SORE impedes the ubiquitination and degradation of Y-box binding protein 1 (YBX1) by binding to the YBX1 protein, thereby enhancing HCC resistance to sorafenib (*Xu et al., 2020*). Interestingly, bioinformatics analysis revealed that circRNAs bind less densely to RBPs than to linear RNAs *via* nucleotide sequence-based binding (*Du et al., 2017*). However, existing studies suggest that the binding of RNA to RBPs is also influenced by the tertiary structure of the RNA molecule. It can be speculated that circRNAs have a more complex binding mechanism to RBPs due to their unique tertiary structure. The mechanism of interaction between circRNAs and RBPs still requires further exploration (*Abe et al., 2015*; *Loughlin et al., 2019*) (Fig. 2C).

### circRNAs as translation templates that encode proteins

Initially, circRNAs were not thought to be able to be translated into proteins. Later, as technological advancements revealed that most circRNAs originate from protein-coding exons in the cytoplasm, circRNAs were thought to have the potential to encode proteins. Currently, some circRNAs have been found to encode proteins both *in vivo* and *in vitro* (*Chen & Sarnow, 1995*). Because circRNAs do not possess a free 5′terminal cap structure, they cannot be translated through the conventional pathway but need to be methylated, specifically *via* N6-methyladenosine (m6A), at an internal ribosome entry site (IRES) or within the 5′ untranslated region (5′ UTR). Methylation of adenosine residues by m6A on the 5′ cap-end initiates protein translation through a 5′ non-cap dependent pathway (*Diallo et al., 2019*; *López-Lastra, Rivas & Barría, 2005*). CircPINT exon 2 is formed by reverse splicing of exon 2 of LINC-PINT, and *Zhang et al. (2018a)* reported that circPINT exon 2 has a natural IRES structure that encodes the PIN87aa peptide, which contains 87 amino acids and has comprehensive tumour-suppressive effects. circ-AKT3 encodes AKT3-174aa, a protein containing 174 amino acids that inhibits the malignant biological behaviour of gliomas and enhances their resistance to radiotherapy when overexpressed (*Xia et al., 2019*). Out of the 32,914 human ecircRNAs selected, 7,170 had IRES elements, indicating their protein-coding potential. Additionally, evidence exists for 37 circRNAs encoding corresponding proteins (*Chen et al., 2016*). Some estimates suggest that at least 13% of all circRNAs have m6A modifications, indicating that a large number of circRNAs could be translated into proteins in ways that have not yet been explored (*Yang et al., 2017*) (Fig. 2D).

### circRNAs can exert regulatory functions outside the cell via the exosome pathway

Exosomes are currently a major area of research and are mainly formed through the process of plasma membrane bilayer invagination and the creation of intracellular multivesicular bodies (MVBs) containing intraluminal vesicles (ILVs). Exosomes are involved in intercellular communication, and their specific roles depend mainly on the bioactives they contain (*Kalluri & LeBleu, 2020*). Recent studies have shown that circRNAs not only play regulatory roles within cells but can also be transported to the extracellular space *via* the exosomal pathway to influence the function of target cells (*Zhang et al., 2022*). circRNAs can regulate tumour proliferation, migration, and invasion *via* exosomes (*Shang et al., 2020*); participate in regulating cellular gene expression and differentiation

*via* exosomes (*Mao et al., 2021*); and contribute to immune function through exosomal regulation. For instance, circNEIL3 can promote macrophage infiltration chemotaxis in the glioma microenvironment (*Pan et al., 2022*). In conclusion, the circRNA-exosome pathway constitutes a highly intricate system that plays crucial biological roles in both intracellular and intercellular communication. Through the study of the association between circRNAs and exosomes, the detection of the composition and content of circRNAs in exosomes can be used as an indicator to predict the stage of disease progression and patient prognosis (Fig. 2E).

## CIRCHECTD1 AND HUMAN-RELATED DISEASES

### The origin of circHECTD1: HECTD1

The circHECTD1 transcript is a mature 344 nt sequence that is transcribed from the HECT domain E3 ubiquitin protein ligase 1 (HECTD1) exons 23–24 on chr14:31602443-31602881(14q12). The circBase database contains the circHECTD1 transcript hsa_circ_0031485, which is highly conserved across species. For instance, in mice, the circHECTD1 transcript is named mmu_circ_0000375 at chr12:52860205-52862157, and its nucleotide sequence is highly similar to that of human circHECTD1 (*Zhou et al., 2018*) (Fig. 3).

HECTD1 is a protein-coding gene located on chromosome 14q12. This gene was first reported to be essential for foetal head mesenchymal development, and mutations in this gene cause foetal head mesenchymal dysplasia resulting in cranial neural tube closure defects (*Zohn, Anderson & Niswander, 2007*). It has also been shown that HECTD1 negatively regulates the Wnt signalling pathway by ubiquitinating adenomatous polyposis coli proteins through interaction with axin to form $\beta$-catenin destruction complexes and phosphorylating $\beta$-catenin proteins (*Tran et al., 2013*). In addition, HECTD1 expression is negatively correlated with mitochondrial respiratory function in breast cancer (*Uemoto et al., 2022*); HECTD1 plays an essential role in the regulation of haematopoietic stem cell function under proliferative stress (*Lv et al., 2021*); and HECTD1 promotes cell proliferation by affecting mitosis, so dysfunction of HECTD1 may be associated with the development of various tumours (*Vaughan et al., 2022*).

Recent studies have demonstrated that circHECTD1 is aberrantly expressed in many human diseases including glioma, gastric cancer, hepatocellular carcinoma, acute ischaemic stroke, silicosis, acute lung injury, ulcerative colitis, atherosclerosis, and hypertrophic scarring. In tumour diseases, circHECTD1 contributes to a variety of malignant biological behaviours including proliferation, migration, invasion, and drug resistance. In nontumour diseases, circHECTD1 has two functions: promoting disease progression in some diseases and alleviating disease progression in others. The following is a review and summary of the role and mechanism of action of circHECTD1 in human diseases, which clarifies the clinical potential of circHECTD1 as a molecular therapeutic target for diseases and a biomarker for prognosis determination and helps to adjust the diagnosis and treatment strategies for various diseases.

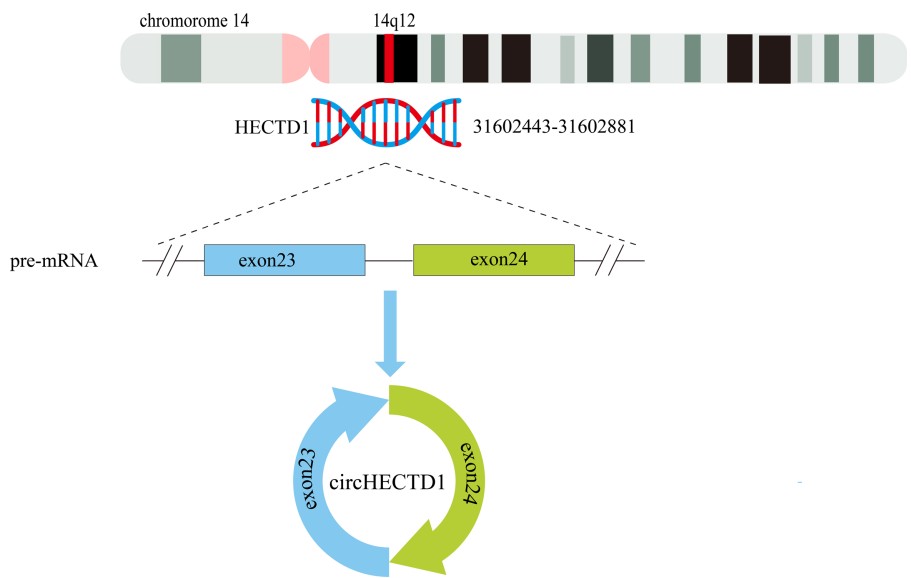

**Figure 3** **Schematic representation of the formation of circHECTD1.** Exons 23–24 of the HECTD1 gene, located on chromosome 14, give rise to a 344-nucleotide circular RNA known as circHECTD1.

## CircHECTD1 and tumour-related diseases
### Glioblastoma

Glioblastoma (GBM) is the most common primary highly aggressive malignant tumour of the central nervous system (*Velásquez et al., 2019*). The prognosis for patients with GBM is often poor, with data showing that the median survival time is only 12–18 months, and the five-year survival rate is only approximately 13.8% after active treatment (*Ghaemi et al., 2022*; *Witthayanuwat et al., 2018*). Therefore, the search for new biomarkers is essential for identifying therapeutic targets and early diagnostic markers that may greatly improve patient prognosis.

Abnormal metabolism is an important cancer marker, and abnormal glycolysis and intracellular signalling play crucial roles in cancer proliferation, migration, and invasion (*Park, Pyun & Park, 2020*). *Li et al. (2021b)* reported that circHECTD1 was highly expressed in GBM cells and that it interacted with miR-320-5p to inhibit its expression. Additionally, solute carrier family 2 member 1 (SLC2A1) was identified as a direct downstream target of miR-320-5p, and its expression was suppressed by miR-320-5p. However, CircHECTD1 upregulates SLC2A1 expression by inhibiting the expression of miR-320-5p. SLC2A1, also known as glucose transporter protein 1 (GLUT1), plays a crucial role in glycolysis and is highly expressed in various cancers, including liver, lung, colorectal, and breast cancers, promoting the proliferation and migration of cancerous cells (*Liu et al., 2022*; *Zheng et al., 2022*).

In addition, *Li et al. (2021a)* discovered that circHECTD1 was highly expressed in GBM tissues but expressed at low levels in normal tissues. CircHECTD1 expression was also positively correlated with the Karnofsky scores of patients, indicating that upregulated circHECTD1 expression is associated with a poor prognosis. Animal experiments

also demonstrated that knockdown of circHECTD1 inhibited GBM growth in mice. Overexpression of circHECTD1 indirectly upregulated solute carrier family 10 member 7 (SLC10A7) expression by suppressing miR-296-3p expression through miRNA sponge action. SLC10A7 is a cytokine that belongs to the solute carrier family SLC10 and regulates intracellular calcium signalling. It also has regulatory effects on various cellular processes, such as cell proliferation, growth, and gene expression (*Berridge, Bootman & Roderick, 2003*; *Durin et al., 2022*; *Karakus et al., 2020*). *Li et al. (2021a)* also demonstrated that overexpression of SLC10A7 could reverse the inhibitory effect of circHECTD1 knockdown on GBM.

Further research has revealed that various circRNAs possess protein-coding functions (*Chen & Sarnow, 1995*). In their most recent study, *Ruan et al. (2023)* reported that a variant of circHECTD1, known as hsa_circ_0002301, has a protein-coding function. The functional peptide 463aa encoded by this gene was identified using LC-MS/MS. The RNA-binding protein RBMS3, which has been implicated in the regulation of various biological processes such as apoptosis, gene transcription, and cell cycle progression (*Penkov et al., 2000*), can specifically bind to the flanking intron sequence of hsa_circ_0002301. This interaction induces the formation of hsa_circ_0002301. *Ruan et al. (2023)* analysed the 463aa peptide sequence and found that it contains a ubiquitination-associated structural domain. This domain promotes the degradation of NR2F1 by regulating the ubiquitination of the K396 site of NR2F1. This site can directly bind to the promoter regions of vasculogenic mimicry (VM)-associated proteins (MMP2, MMP9, and VE-cadherin), thereby enhancing GBM VM. Furthermore, hsa_circ_0002301 and 463aa significantly inhibited the proliferation, migration, invasion, and tubular structure formation of GBM cells, as well as VM formation *in vitro* and in vivo. This study sheds light on a novel pathway in which the RBMS3-induced circHECTD1-encoded functional peptide 463aa mediates the ubiquitination of NR2F1, thereby inhibiting VM in GBM.

In conclusion, circHECTD1 can lead to abnormal cell metabolism and promote the development of GBM through the miR-320-5p/SLC2A1 and miR-296-3p/SLC10A7 axes. Conversely, the variant hsa_circ_0002301 encodes a functional peptide 463aa, which inhibits the proliferation, migration, invasion, tube formation, and VM of GBM *in vitro* and *in vivo*, thereby improving the prognosis of patients with GBM. These studies indicate that circHECTD1 plays a crucial regulatory role in GBM and provides potential targets for developing new treatment strategies. However, these findings need to be validated in larger samples and further research is required to determine the clinical application value of these discoveries.

### Gastric carcinoma

Gastric cancer (GC) is a prevalent malignancy of the digestive system and ranks among the top five most commonly diagnosed cancers worldwide. Its estimated annual incidence surpasses 1 million new cases (*Smyth et al., 2020*). Nevertheless, GC has become a significant global health issue due to its high mortality rate attributable to late detection, tumour drug resistance, and other factors (*Petryszyn, Chapelle & Matysiak-Budnik, 2020*; *Schinzari et al.,*

*2014*). Hence, exploring novel biomarkers and investigating mechanisms of drug resistance could offer new strategies and targets for treating GC.

Drug resistance is a significant hurdle in cancer treatment (*Wu et al., 2014*). Studies have reported the association of circRNAs with the development of drug resistance in cancer (*Chen et al., 2021*). Diosbulbin-B (DB), a diterpenoid with antitumour activity, is hepatotoxic at high concentrations (*Wang et al., 2014*). Therefore, increasing tumour sensitivity to DB can effectively improve its therapeutic effect while reducing liver damage. To address this issue, *Lu et al. (2021)* found that inhibiting circHECTD1 led to restricted tumour growth, an increased proportion of GC cells stalled in the G0/G1 phase by DB and a significant decrease in the median lethal dose (LD50) of DB in GC cells. circHECTD1 indirectly upregulates pre-B-cell leukaemia homeobox 3 (PBX3) *via* competitive inhibition of miR-137, and overexpression of PBX3 increases drug resistance in gastric cancer cells to DB. PBX3 belongs to the pre-B-cell leukaemia family and has been reported to play crucial roles in various cancers such as acute myeloid leukaemia, gastric cancer, colorectal cancer, and hepatocellular carcinoma (*Morgan & Pandha, 2020*). *Cai et al. (2019)* conducted a study that revealed significantly higher expression levels of circHECTD1 in gastric cancer tissues than in normal tissues. Multivariate survival analysis revealed that patients with high circHECTD1 expression had significantly shorter overall survival than those with low expression, indicating that increased circHECTD1 expression is linked to poor prognosis. Moreover, overexpression of circHECTD1 inhibited miR-1256, leading to the upregulation of USP5 and activation of the $\beta$-catenin/c-MYC signalling pathway, thus promoting glutamine catabolism. Conversely, knockdown of circHECTD1 reduced $\beta$-catenin/c-MYC signalling pathway activation and decreased glutamine catabolism. The c-MYC gene family plays a crucial role in regulating the expression of thousands of genes directly or indirectly, and its products are frequently found to be activators in human cancers (*Dhanasekaran et al., 2022*). In contrast, a high rate of glutamine consumption is a common metabolic feature of cancer (*Márquez et al., 2017*).

While *in vitro* experiments have demonstrated the ability of circHECTD1 to enhance the resistance of GC cells to DB through the miR-137/PBX3 pathway and promote cancer development *via* the miR-1256/USP5 pathway, further *in vivo* experiments are necessary to confirm its impact on DB resistance. Inhibiting circHECTD1 not only enhances chemotherapy efficacy but also reduces cancer cell proliferation and metabolism, indicating its potential as a future therapeutic target.

### Hepatocellular carcinoma

Hepatocellular carcinoma (HCC) is a type of primary liver cancer that constitutes approximately 90% of such cases (*Dimitroulis et al., 2017*; *Forner, Reig & Bruix, 2018*). According to global cancer statistics, HCC ranks as the fourth most common cancer and the sixth leading cause of cancer-related deaths (*Villanueva, 2019*). HCC is known for its aggressive nature and high susceptibility to metastasis. Unfortunately, it is often identified in later stages due to a lack of specific early symptoms and effective diagnostic markers, which leads to a poor prognosis (*Ge & Huang, 2015*). Thus, the identification of potential biomarkers for HCC is critical for improving the diagnosis and treatment of this disease.

Numerous studies have demonstrated that circRNAs are involved in regulating various malignant biological processes in HCC by utilizing a diverse array of pathways, including competitive inhibition of endogenous RNA (*Li et al., 2020a*), binding to RNA-binding proteins (*Liu et al., 2020*) and exosomal pathways (*Huang et al., 2020*). For instance, a study conducted by *Jiang et al. (2020)* revealed that circHECTD1 upregulated MUC1 through competitive inhibition of miR-485-5p. Subsequent qRT-PCR analyses confirmed that MUC1 expression was notably greater in HCC tissues than in normal tissues. Additionally, Kaplan–Meier survival curves suggested that patients with high MUC1 expression exhibited substantially shorter survival times than those with low MUC1 expression, suggesting that elevated MUC1 expression indicates a poor prognosis for individuals diagnosed with HCC. Moreover, recent studies have indicated that miR-485-5p functions as an oncogene, playing significant roles in the pathogenesis of multiple types of cancer such as glioma (*Cheng et al., 2021*), hepatocellular carcinoma (*Zhao et al., 2022*), and lung adenocarcinoma (*Chen, Wu & Bao, 2023*), and is also implicated in disease progression associated with distinct degenerative disorders, such as osteoporosis (*Zhang et al., 2018b*) and Alzheimer's disease (*He et al., 2021b*).

It is evident that circHECTD1 plays a role in the pathophysiology of hepatocellular carcinoma by regulating the miR-485-5p/MUC1 network. Although the lack of *in vivo* experiments limits the conclusions of the study, it cannot be denied that this represents a promising therapeutic target and prognostic indicator for HCC. Furthermore, the observed inhibitory effect of circHECTD1 on miR-485-5p suggests its potential involvement in the pathogenesis of other cancers and the modulation of certain degenerative diseases in humans. However, due to current limitations in research, further investigation is necessary, with immense potential for future studies.

## CircHECTD1 and nontumour-related diseases
### Acute ischaemic stroke

Acute ischaemic stroke (AIS) is one of the major types of strokes. AIS is more common than haemorrhagic stroke and can cause significant disability or even death in patients (*Campbell et al., 2019*). The goal of treatment is to prevent irreversible nerve damage. Currently, recombinant tissue-type plasminogen activator (rtPA) reperfusion is the only FDA-approved treatment modality for acute ischaemic stroke. While new treatment modalities such as stem cell therapy are being explored, effective options remain limited (*Barthels & Das, 2020*; *Paul & Candelario-Jalil, 2021*). Many patients therefore endure long-term disability because they do not receive optimal treatment within the appropriate time frame (*Li, Tang & Yang, 2021d*). As a result, exploring new therapeutic targets and diagnostic markers is essential. *Peng et al. (2019)* conducted a study comparing patients with AIS with normal controls and found a positive correlation between circHECTD1 expression and disease severity in patients with AIS. Additionally, Kaplan–Meier survival curve analysis revealed that patients with high circHECTD1 expression had significantly shorter survival than those with low circHECTD1 expression. Patients with high circHECTD1 expression also had a significantly greater probability of recurrence than did those with low circHECTD1 expression. These findings suggest that increased expression of

circHECTD1 is a poor prognostic indicator for patients with AIS. Additionally, *Han et al. (2018)* reported that circHECTD1 expression was significantly greater in the plasma of patients with AIS than in that of normal controls. They also found that knockdown of circHECTD1 reduced the extent of brain infarct injury in mice. In addition, they observed decreased expression of glial fibrillary acidic protein (GFAP), which suggests that circHECTD1 is involved in astrocyte activation. Astrocytes are been considered passive cells that provide support to neurons, but in recent years, astrocytes have been shown to play an important role in nerve injury and repair (*Bugiani et al., 2022*). *Han et al. (2018)* demonstrated by FISH experiments and immunofluorescence assays that circHECTD1 competitively inhibits miR-142 upregulation of TIPARP to promote astrocyte autophagy, which in turn promotes astrocyte activation. *Zhang, He & Wang (2021)* reported that the expression of circHECTD1 was elevated in HT22 cells after treatment with oxygen–glucose deprivation/reperfusion (OGD-R). circHECTD1 upregulates follistatin-like 1 (FSTL1) by acting as a sponge for miR-27a-3p, which is involved in OGD-R-induced neuronal cell injury. Many studies have demonstrated that FSTL1 participates in tissue remodelling and inflammatory processes, mediating cell–matrix interface interactions and contributing to various fibrotic and autoimmune diseases (*Li et al., 2021c*). Conversely, *He et al. (2022)* discovered that circHECTD1 acted as a sponge for miR-355 to indirectly upregulate notch receptor 2(NOTCH2) and promote endothelial-mesenchymal transition (EndMT) in human cerebral microvascular endothelial cells (HCMECs). Knocking down CircHECTD1 inhibited EndMT and facilitated migration and tube-formation in OGD-R-treated HCMECs.

Studies suggest that circHECTD1 may play roles in astrogliosis and EndMT in AIS through the miR-142/TIPARP and miR-355/NOTCH2 axes, respectively. It can also contribute to neuronal cell injury *via* the miR-27a-3p/FSTL1 axis. The primary source of experimental research samples was patient plasma, which may not directly reflect changes in the site of the lesion, leading to potential bias in the results. The reliability of the experiment can be further proven by examining the patient's cerebrospinal fluid. However, peripheral blood can indirectly reflect changes at lesion sites, demonstrating the significant potential of circHECTD1 as a biomarker for evaluating disease severity and prognosis.

### Silicosis

Silicosis is a pulmonary fibrosis disease that is mainly caused by progressive fibrotic changes in the lungs as a result of long-term exposure to an environment containing high levels of free silica dust. A serious and common occupational disease (*Leung, Yu & Chen, 2012*; *Steenland & Ward, 2014*), Silicosis starts with the phagocytosis of silica crystals by alveolar macrophages (*Tan & Chen, 2021*). Subsequently, silica-induced apoptosis of macrophages triggers a chain reaction of inflammatory responses that ultimately induces fibroblast activation and leads to lung fibrosis (*Hu et al., 2006*). Currently, there is a lack of effective diagnostic methods for detecting silicosis before it progresses to an advanced stage (*Austin, James & Tessier, 2021*). There are also only a limited number of drugs available to slow disease progression, and therapeutic options remain ineffective (*Adamcakova & Mokra,*

*2021*; *Wang et al., 2022*). As a result, there is an urgent need to identify potential therapeutic targets for silicosis to develop effective treatments for this disease.

According to a study by *Zhou et al. (2018)*, circHECTD1 was highly expressed in RAW264.7 cells exposed to SiO2. Furthermore, circHECTD1 regulated the ubiquitination of HECTD1/ZC3H12A, which resulted in the activation of alveolar macrophages. Additionally, exposure to SiO2 upregulated HECTD1, which promoted the activation and migration of human pulmonary fibroblasts from adults (HPF-a). *Chu et al. (2019)* conducted further research into the relationship between circHECTD1/HECTD1 and HPF-a, building upon the study by *Zhou et al. (2018)*. As per their findings, under SiO2 exposure, circHECTD1 indirectly promoted the proliferation and migratory capacity of HPF-a, reduced apoptosis, and promoted cellular autophagy through the upregulation of HECTD1 expression in both HPF-a and patient cells. Numerous studies have indicated that cellular autophagy involves not only a "garbage can" but also a dynamic recycling system. In some instances, autophagy can function as an energy generator. It has also been demonstrated that autophagy is linked to the activation of HPF-a migration (*Mizushima & Komatsu, 2011*; *Zhu et al., 2016*). *Fang et al. (2018)* reported contrasting findings that SiO2 exposure elevated circHECTD1 expression while suppressing HECTD1 protein expression, and promoted EndMT in mouse endothelial cells. It has been demonstrated that epithelial–mesenchymal transition (EMT) and EndMT in epithelial and endothelial cells, caused by various factors, are significant contributors to fibroblast and extracellular matrix (ECM) accumulation leading to the development of fibrotic disease (*Wynn, 2008*).

In conclusion, circHECTD1 has dual roles in the development of pulmonary fibrosis. While it indirectly promotes the activation of AMOs and modulates the function of HPF-a by upregulating the HECTD1/ZC3H12A axis, it can also promote EndMT in lung endothelial cells and contribute to lung fibrosis. These studies did not further investigate the specific mechanism by which circHECTD1 promotes EndMT. Subsequent research can build upon this foundation, but it is undeniable that circHECTD1 plays a crucial regulatory role in promoting pulmonary fibrosis.

### Acute lung injury

Acute lung injury (ALI), also known as mild or moderate acute respiratory distress syndrome (ARDS), is a disease characterized by diffuse alveolar injury with disruption of the epithelial-endothelial barrier caused by direct or indirect injurious factors (*Butt, Kurdowska & Allen, 2016*). ALI is associated with high morbidity and mortality (*Mowery, Terzian & Nelson, 2020*). Currently, nonpharmacological treatment for ALI mainly involves lung-protective mechanical ventilation as there are no specific drug therapies available (*Mokrá, 2020*). Therefore, there is an urgent need to explore molecularly targeted therapeutic agents for ALI to reduce its high morbidity and mortality.

As research on molecular pathophysiology has advanced, increasing evidence suggests that circRNAs play a significant regulatory role in the repair of damage in ALI (*Cao et al., 2022*). According to a study by *Li et al. (2022a)*, there was a marked reduction in the expression of circHECTD1 in human and mouse alveolar epithelial cells (AECs) stimulated with lipopolysaccharide (LPS). However, the overexpression of circHECTD1 has been

found to upregulate phosphatidylinositol-4,5-bisphosphate 3-kinase catalytic subunit alpha (PIK3CA) and sirtuin 1 (SIRT1) indirectly by acting as a sponge for miR-320a and miR-136. Notably PIK3CA is involved in numerous cellular processes such as angiogenesis, survival, metabolism, and proliferation (*Canaud et al., 2021*), while SIRT1 is a highly conserved NAD-dependent deacetylase with anti-inflammatory and antioxidant properties that can help mitigate tissue damage caused by inflammation (*Yang et al., 2022*). (*Li et al., 2022a*) discovered that overexpression of circHECTD1 inhibited the programmed cell death of AECs.

In summary, circHECTD1 can activate both the miR-320a/PIK3CA and miR-136/SIRT1 pathways in LPS-induced ALI, promoting cell proliferation and metabolism while also preventing further cell damage caused by inflammation and slowing AECs apoptosis. A thorough study of the mechanisms underlying the effects of circHECTD1 on ALI could be tremendously helpful in understanding and preventing the progression of ALI to ARDS, as well as developing targeted therapies for ALI.

### *Ulcerative colitis*

Ulcerative colitis (UC) is a diffuse and chronic inflammatory disease that primarily affects the superficial mucosa of the rectum and colon, causing mucous bleeding. UC is characterized by various clinical manifestations, with acute UC episodes potentially leading to toxic megacolonitis, while long-term chronic UC may increase the risk of cancer development (*Danese & Fiocchi, 2011*; *Ordás et al., 2012*). Unfortunately, there are currently no drugs that can completely cure UC (*Wehkamp & Stange, 2018*). The primary objective of treating UC is to prevent its recurrence and induce remission of active UC (*Adams & Bornemann, 2013*). Furthermore, the global incidence and prevalence of UC have shown increasing trends over the years (*Ungaro et al., 2017*). Therefore, it is crucial and pressing to investigate the underlying causes of UC and uncover new potential therapeutic targets.

Currently, with the progress of molecular biology research, there is growing evidence suggesting that circRNAs play a role in the progression and regression of UC *via* miRNA/mRNA networks (*Xu et al., 2022a*). *Xu et al. (2022b)* reported that the expression of circHECTD1 was lower in UC tissues than in normal tissues. They also found that circHECTD1 could indirectly upregulate human antigen R (HuR) by acting as an miRNA sponge, which reduced the secretion of inflammatory factors such as IL-1 $\alpha$, IL-1 $\beta$, and IL-6, and increased the expression of autophagy-related genes such as ATG5 and ATG9. These findings suggested that circHECTD1 may mitigate the inflammatory response in UC through cellular autophagy. Interestingly, many studies have demonstrated that circRNAs often act as oncogenes in cancer, promoting the development of various types of cancer (*Sang et al., 2019*). However, in UC, circRNAs act by promoting cellular autophagy to alleviate the inflammatory response associated with the disease (*Xu et al., 2022b*).

Indeed, the aetiology of UC remains unclear and may result from various factors, including genetics, immune abnormalities, and dysbiosis of the intestinal flora (*Du & Ha, 2020*). The relationship between reduced circHECTD1 expression and UC has provided new insights for investigating the pathogenesis of this disease. *In vivo* and *in vitro* experiments

have demonstrated that circHECTD1 can effectively alleviate the development of UC. However, these findings require validation in larger samples and further research to determine the clinical utility of these discoveries.

### Atherosclerosis

Cardiovascular disease is responsible for more global deaths than any other disease, making it the number one cause of mortality worldwide. Atherosclerosis (AS) is a significant pathological change that can lead to cardiovascular disease (*Herrington et al., 2016*). Numerous studies have demonstrated that the collagen produced by leukocytes and vascular smooth muscle cells (VSMCs) as they move towards the intima serves as the primary origin of the fibrous cap (FC) and plays a crucial role in preserving plaque stability (*Miano, Fisher & Majesky, 2021*). Autophagy and apoptosis of VSMCs can adversely impact the stability of the FC and contribute to the formation of necrotic foci, ultimately resulting in plaque rupture (*Grootaert et al., 2018*). Consequently, identifying a biological target that can enhance VSMC proliferation and reduce VSMC apoptosis is crucial. This may serve as a significant therapeutic approach for treating and slowing the progression of AS.

A study by *Feng et al. (2023)* revealed that the expression of circHECTD1 was elevated in VSMCs stimulated with platelet-derived growth factor-BB (PDGF-BB). They found that circHECTD1 enhanced the stability of EZH2 mRNA and increased the expression of EZH2 protein through an interaction with the RNA-binding protein KHDRBS3. This mechanism promoted VSMC proliferation and migration while inhibiting apoptosis. Indeed, elevated levels of EZH2 can disrupt the normal patterns of histone modifications, resulting in abnormal epigenetic regulation and contributing to tumour development (*Hanaki & Shimada, 2021*). Breast cancer (*Li et al., 2020b*), gastric cancer (*Pan et al., 2016*), prostate cancer (*Park et al., 2021*), and numerous other types of cancer are specifically associated with these aberrations caused by overexpressed EZH2. In summary, circHECTD1 enhances the proliferation and inhibits the apoptosis of VSMCs *via* the KHDRBS3/EZH2 axis. Consequently, it represents a potential biomarker for assessing prognosis and treating AS. However, it is important to note that circHECTD1 has dual effects. While EZH2 increases expression, which delays AS progression, the overexpression of EZH2 itself has been associated with cancer development.

### Myocardial ischaemia–reperfusion injury

When acute myocardial infarction (AMI) occurs, early revascularization of the infarct-related coronary artery is important for preventing further deterioration of the AMI. However, during reperfusion therapy, there is a risk of myocardial ischaemia-reperfusion injury (MIRI), which can result in secondary damage to the heart (*Samsky et al., 2021*; *Zhao et al., 2021*). Previous studies have indicated that factors such as inflammation, the release of reactive oxygen species, and local microvascular occlusion play crucial roles in the development of MIRI (*Li, Sun & Li, 2019b*). Therefore, targeted interventions aimed at controlling inflammation and reducing the release of reactive oxygen species can effectively decrease the likelihood of MIRI during reperfusion. Recent studies have demonstrated the critical regulatory roles of ncRNAs in cardiovascular diseases, including MIRI (*Marinescu et al., 2022*). In a study conducted by *Yang et al. (2023)*, they observed

heightened expression of a specific ncRNA known as circHECTD1 in a mouse model of MIRI, particularly in ischaemic tissues. Their findings indicated that circHECTD1 increased the expression of ROCK2 in cells by functioning as an miR-138-5p sponge, consequently fostering inflammatory responses. ROCK2 is a serine/threonine kinase, and research has indicated that increased levels of ROCK2 in cardiovascular diseases can lead to oxidative stress and enhance inflammatory responses, thus exacerbating cardiovascular injury (*Shimokawa, Sunamura & Satoh, 2016*). In the MIRI model, downregulation of circHECTD1 resulted in decreased expression of ROCK2, as well as reduced production and release of proinflammatory factors such as TNF-$\alpha$, IL-6, and IL-1$\beta$ (*Yang et al., 2023*).

Based on these findings, it can be concluded that circHECTD1 plays a role in regulating the cellular inflammatory response through the miR-138-5p/ROCK2 axis in the MIRI model, contributing to the development and exacerbation of MIRI. Treatment targeting circHECTD1 holds promise for reducing the occurrence of MIRI and preventing secondary damage in patients with AMI, potentially improving overall treatment outcomes. However, further research and clinical studies are needed to fully explore the therapeutic potential of circHECTD1 in the context of MIRI.

### Hypertrophic scars

Hypertrophic scars (HS) are characterized by an excessive and prolonged response to dermal injury, resulting in fibrous tissue overgrowth. They typically occur as a result of damage to the reticular dermis and aberrant wound healing processes. Various factors can contribute to the development of HS, including trauma, burns, surgical incisions, vaccinations, skin piercings, acne, and herpes zoster infections (*Lee & Jang, 2018*; *Ogawa, 2017*). Previous studies have demonstrated that persistent activation of the TGF-$\beta$/SMAD signalling pathway is a significant factor in abnormal wound healing and scars formation (*Zhang et al., 2020*). Therefore, investigating biomolecular markers that regulate the excessive activation of the TGF-$\beta$/SMAD signalling pathway could help reduce scar formation.

Recent studies in the field of epigenetics have revealed that aberrant circRNA expression has a reversible impact on various biological behaviours of fibroblasts by affecting epigenetic mechanisms (*Lv et al., 2020*). *Ge et al. (2022)* reported significantly elevated levels of circHECTD1 expression in scar tissue compared to normal tissue. The presence of circHECTD1 indirectly enhances the TGF-$\beta$/SMAD signalling pathway through the miR-142-3p/HMGB1 pathway, resulting in increased human skin fibroblast (HSF) proliferation, migration, and fibrosis. These processes subsequently contribute to abnormal wound healing and scar formation. Recent studies have indicated that increased expression of transforming growth factor $\beta$ (TGF-$\beta$) can result in metabolic and functional impairments while also promoting the progression of epithelial–mesenchymal transition (EMT) and excessive extracellular matrix (ECM) deposition. These effects ultimately contribute to the development of fibrosis (*Peng et al., 2022*). SMAD transcription factors function as crucial mediators of TGF-$\beta$ signalling. Moreover, according to existing studies, it is suggested that the phosphorylation of the SMAD-linked region itself serves as a signalling pathway (*Kamato et al., 2020*).

In summary, circHECTD1 promotes HS formation primarily by activating the TGF-$\beta$/SMAD signalling pathway through the miR-142-3p/HMGB1 axis. However, the sample size of the study was small, which may have led to bias. Therefore, the generalizability of these findings needs to be validated in a larger sample.

## THE ROLES OF CIRCHECTD1 IN DIFFERENT DISEASES

With the advancement of research on circRNAs, circHECTD1, a new molecular target, has been increasingly explored. Numerous studies have shown that CircHECTD1 is aberrantly expressed in a range of diseases, including glioblastoma, gastric cancer, hepatocellular carcinoma, ischaemic stroke, silicosis, acute lung injury, inflammatory bowel disease, and hypertrophic scars. CircHECTD1 influences the progression of these diseases in various ways, such as disease onset, prognosis, and drug resistance. The role of circHECTD1 varies across different diseases, affecting aspects such as disease onset, prognosis, and drug resistance (Fig. 4).

In diseases related to tumours, circHECTD1 indirectly enhances the expression of miRNA downstream target genes, primarily through miRNA sponging. This promotes various malignant biological behaviours, such as the proliferation, migration, and invasion of cancer cells. For instance, circHECTD1 increases the expression of SLC2A1 by inhibiting miR-320-5p, and increases the expression of SLC10A7 by associating with miR-396-3p. These actions contribute to the development of GBM by causing abnormal cellular metabolism (*Li et al., 2021a*; *Li et al., 2021b*). CircHECTD1 also promotes the development of HCC through the miR-485-5p/MUC1 pathway (*Jiang et al., 2020*). In gastric cancer, circHECTD1 plays a dual role. It not only activates the $\beta$-catenin/c-Myc signalling pathway *via* the miR-1256/USP5 pathway, thereby promoting cancer progression but also increases the resistance of GC cells to DB by upregulating PBX3 through its interaction with miR-137 (*Cai et al., 2019*; *Lu et al., 2021*). Consequently, high expression of circHECTD1 often indicates poor patient prognosis. In the latest study, a variant of circHECTD1, hsa_circ_0002301, was found to have protein-coding functions. The functional peptide 463aa encodes a gene that significantly inhibits the proliferation, migration, invasion, and tubular structure formation of GBM cells, as well as VM formation, both *in vitro* and *in vivo* (*Ruan et al., 2023*). These variants exhibit oncogene suppression, which contradicts the findings of previous studies indicating that circHECTD1 is a proto-oncogene. According to data from the circBase database, all circRNA variants generated by HECTD1 pre-mRNA can produce different circHECTD1 variants based on the number of their circularized exons or introns, leading to changes in their functions.

In nontumour diseases, the role of circHECTD1 is diverse. In conditions such as acute ischaemic stroke, silicosis, and hypertrophic scarring, circHECTD1 contributes to disease progression by promoting the proliferation of associated injured cells, as well as EMT and EndMT. For instance, in acute ischaemic stroke, circHECTD1 influences astrocyte activation and EndMT through the miR-142/TIPARP and miR-355/NOTCH2 axes, respectively (*Han et al., 2018*; *Zhang, He & Wang, 2021*). In silicosis, circHECTD1 promotes EndMT in lung endothelial cells *via* upregulation of the HECTD1/ZC3H2A axis,

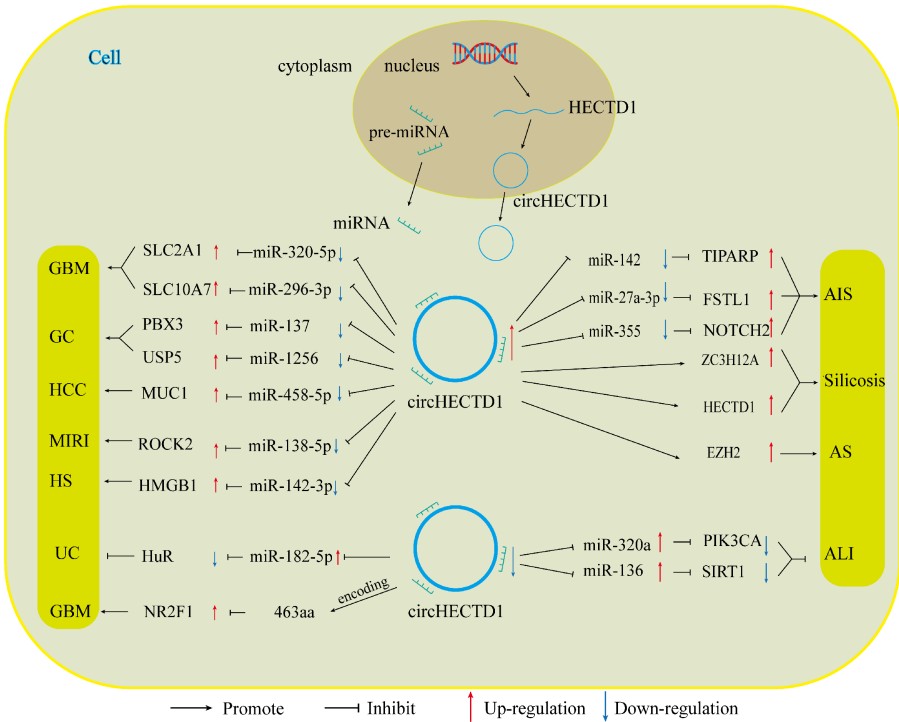

**Figure 4 CircHECTD1 participates in various diseases through different ways.** The up-regulation of circHECTD1 is associated with the occurrence and development of GBM, GC, HCC, AIS, silicosis, MIRI, and HS. Conversely, the down-regulation of circHECTD1 is linked to the occurrence and development of UC and ALI.

thereby contributing to lung fibrosis (*Chu et al., 2019*; *Fang et al., 2018*). Conversely, in conditions such as acute lung injury, inflammatory bowel disease, myocardial ischaemia-reperfusion injury, and atherosclerosis, circHECTD1 plays protective roles, inhibiting inflammation and promoting the proliferation of related repair cells to prevent further tissue damage. This is achieved by competitively inhibiting miRNAs (*e.g.*, miR-320a/PIK3CA, miR-136/Sirt, miR-182-5p/HuR, and miR-138-5p/ROCK2) (*Li et al., 2022a*; *Xu et al., 2022b*; *Yang et al., 2023*). This helps to prevent further tissue damage caused by inflammatory factors, thereby alleviating disease progression (Table 1).

These differential effects underscore the complex and context-dependent nature of circHECTD1 in different diseases.

## CONCLUSIONS AND FUTURE PERSPECTIVES

CircRNAs play crucial roles in the development of diverse diseases. The advancement of single-cell transcriptomics and high-throughput sequencing technologies has led to the exploration of circRNAs, which are a type of ncRNA. This discovery has revealed a novel perspective in the field of RNA molecule research. The biological properties and functions of circRNAs are gradually being discovered, helping us to gain insight into the mechanisms of action of circRNAs. Studies of circRNAs hold great promise, particularly in the realms

**Table 1  Expression regulation and roles of circHECTD1 in various diseases including malignant tumours.**

| Disease types | Cell lines | Expression | Related miRNA | Related proteins | Functional roles | References |
|---|---|---|---|---|---|---|
| Glioblastoma | C6, U87, HEK-293T | Up-regulation | miR-320-5p | SLC2A1 | Promoting cell proliferation, migration, invasion and glycolysis | *Li et al. (2021b)* |
| | T98G, SHG44, U251, LN229, A172, NHA | Up-regulation | miR-296-3p | SLC10A7 | Promoting cell proliferation, migration and invasion | *Li et al. (2021a)* |
| | U87, U251 | Down-regulation | – | 436aa, NR2F1 | Encodes functional peptide 463aa; Ubiquitination of NR2F1, inhibition of vasculogenic mimicry | *Ruan et al. (2023)* |
| Gastric Carcinoma | AGS, HGC-27 | Up-regulation | miR-137 | PBX3 | Induced cell cycle and suppressing cell apoptosis; Increasing drug resistance | *Lu et al. (2021)* |
| | BGC823, MKN45, HGC27, AGS, MGC803, SGC7901, GES-1 | Up-regulation | miR-1256 | USP5 | Promoting cell proliferation, migration, invasion and cellular metabolism | *Cai et al. (2019)* |
| Hepatocellular Carcinoma | THLE-3, HCCLM, MHCC97L, SMMC7721-1640, Hep3B, HepG2 | Up-regulation | miR-485-5P | MUC1 | Promoting cell proliferation, migration and invasion; suppressing cell apoptosis | *Jiang et al. (2020)* |
| Acute Ischemic Stroke | A172 | Up-regulation | miR-142 | TIPARP | Promotes astrocyte autophagy and activation | *Han et al. (2018)* |
| | HT22 | Up-regulation | miR-27a-3p | FSTL1 | Suppressing cell proliferation; promoting cell apoptosis | *Zhang, He & Wang (2021)* |
| | HCMEC | Up-regulation | miR-355 | NOTCH2 | Increasing EndMT and facilitates migration and tube formation of HCMECs. | *He et al. (2022)* |
| Silicosis | RAW264.7, L929, | Up-regulation | – | HECTD1, ZC3H12A | Promoting cell proliferation, migration | *Zhou et al. (2018)* |
| | HPF-a | Up-regulation | – | HECTD1 | Induced cell activation, proliferation, migration | *Chu et al. (2019)* |
| | MML2 | Up-regulation | – | HECTD1 | Induced EndMT | *Fang et al. (2018)* |

**Table 1** (*continued*)

| Disease types | Cell lines | Expression | Related miRNA | Related proteins | Functional roles | References |
|---|---|---|---|---|---|---|
| Acute Lung Injury | HBE, MLE-12 | Down-regulation | miR-320a; miR-136 | PIK3CA; SIRT1 | Suppressing cell apoptosis | *Li et al. (2022a)* |
| Ulcerative colitis | Caco-2 | Down-regulation | miR-182–5p | HuR | Promotes cellular autophagy | *Xu et al. (2022b)* |
| Atherosclerosis | HBVSMC | Up-regulation | – | KHDRBS3, EZH2 | Promoting cell proliferation, migration; suppressing cell apoptosis | *Feng et al. (2023)* |
| Myocardial ischemia–reperfusion injury | H9c2 | Up-regulation | miR-138-5p | ROCK2 | Promotes cellular inflammation | *Yang et al. (2023)* |
| Hypertrophic scar | HDF, HSF | Up-regulation | miR-142-3p | HMGB1 | Promotes cell proliferation, migration, invasion, fibrosis and TGF-β/Smad signaling. | *Ge et al. (2022)* |

of RNA biology and disease treatment. Continued investigations into circRNAs in the domains of gene regulation and disease therapy have the potential to yield novel prognostic indicators for medicine and facilitate the advancement of circRNA-based pharmaceuticals.

CircHECTD1 has emerged as a novel molecular target through extensive research on circRNAs. Numerous studies have highlighted its dysregulated expression in various diseases, such as glioblastoma, gastric cancer, hepatocellular carcinoma, ischaemic stroke, silicosis, acute lung injury, ulcerative colitis and hypertrophic scarring. The aberrant expression of circHECTD1 affects disease progression by influencing factors such as disease onset, prognosis, and drug resistance. This highlights its potential as a valuable biomarker and therapeutic target in these conditions.

The aforementioned studies also indicate that circHECTD1 is widely distributed in human tissues. However, there is a lack of research on the expression and mechanism of action of circHECTD1 in other tissue systems. Additionally, the current understanding of the mechanism of action of circHECTD1 is not comprehensive. Most studies have focused on its role as an miRNA sponge, while other functional mechanisms of circHECTD1 have not been extensively explored.

For instance, there is a lack of research reports on the molecular regulatory mechanisms upstream and downstream of circHECTD1, the role of circHECTD1 in interacting with RBPs to regulate protein function, and the regulation of transcription and translation of related mRNAs. Studies have shown that circRNAs can encode proteins *via* open reading frames (ORFs) and IRES (*Diallo et al., 2019*; *López-Lastra, Rivas & Barría, 2005*). Analysis of the CircRNADb database revealed that circHECTD1 contains ORF and IRES components, and (*Ruan et al., 2023*) study confirmed that circHECTD1 has a protein-coding function. However, studies concerning the protein-coding function of circHECTD1 are rare. Studying the functional structure of circHECTD1 in depth can indeed lead to a better understanding of its important roles in living organisms. For example, it has been shown that many IRES-like elements can act as trans-acting factors to drive the translation of circRNAs

(*Fan et al., 2022*). Additionally, cis-acting RNA motifs play a crucial role in determining the biogenesis and functions of circRNAs (*Dong et al., 2017*). It is possible to design a controllable circHECTD1 translation tool using cis or trans-acting factors, which can promote the production of more therapeutic proteins within organisms. This provides a new molecular-level treatment approach to improve the progression and prognosis of patients' diseases. This deeper understanding can shed light on its mechanisms of action in various life science fields and provide insights into its involvement in disease processes at the molecular level. By revealing the intricate mechanisms underlying the functions of circHECTD1, it is possible to generate new ideas for diagnosing and treating diseases. Additionally, this research may identify novel targets for drug development, leading to the creation of more effective and targeted medications that can improve patient outcomes.

Furthermore, EMT and EndMT are among the current hot research topics and have significant implications for fibrosis. Studies on acute ischaemic stroke, silicosis, and proliferative scarring have shown that circHECTD1 plays a crucial role in fibrosis development. Further in-depth investigations into the functional mechanisms of circHECTD1 can greatly enhance our understanding of fibrotic diseases at the molecular level. This understanding can aid in the development of innovative treatment approaches for fibrosis and the exploration of targeted drugs, ultimately improving the prognosis of patients with fibrotic conditions.

In summary, circHECTD1, a recently discovered circular RNA, has garnered increasing attention in studies elucidating its mechanisms and roles in various diseases. These findings strongly suggest the potential of circHECTD1 as both a biomarker and a therapeutic target for diseases. Furthermore, numerous functions of circHECTD1 remain unexplored, highlighting significant opportunities for further investigation and exploration.

## ACKNOWLEDGEMENTS

The authors thank the current members of the laboratory.

### Funding

This article was supported by the National Natural Science Fund, grant number 82073090; Shanxi Provincial Department of Human Resources and Social Security, grant number 20210001. The funders had no role in study design, data collection and analysis, decision to publish, or preparation of the manuscript.

### Grant Disclosures

The following grant information was disclosed by the authors:
National Natural Science Fund: 82073090.
Shanxi Provincial Department of Human Resources and Social Security: 20210001.

### Competing Interests

The authors declare there are no competing interests.

## Author Contributions

- Yiran Yuan conceived and designed the experiments, performed the experiments, analyzed the data, prepared figures and/or tables, and approved the final draft.
- Xiaomin Zhang performed the experiments, analyzed the data, prepared figures and/or tables, and approved the final draft.
- Xiaoxiao Wang conceived and designed the experiments, analyzed the data, prepared figures and/or tables, and approved the final draft.
- Lei Zhang analyzed the data, authored or reviewed drafts of the article, and approved the final draft.
- Jiefeng He analyzed the data, authored or reviewed drafts of the article, and approved the final draft.

## Data Availability

This study is a literature review.

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
