# Peer review of "The emerging roles of circHECTD1 in human diseases and the specific underlying regulatory mechanisms"

_PeerJ, doi:10.7717/peerj.17612_

## Round 0.1 · original submission · Major Revisions

Please address the critiques of both reviewers and revise the manuscript accordingly.

**Language Note:** The review process has identified that the English language must be improved. PeerJ can provide language editing services - please contact us at [email protected] for pricing (be sure to provide your manuscript number and title). Alternatively, you should make your own arrangements to improve the language quality and provide details in your response letter. – PeerJ Staff

Reviewer 1 ·

Basic reporting

Circ RNA, an important type of non-coding RNA, is known to play significant roles in many important diseases such as cancer. CircHECTD1 is one of these circRNAs, and a detailed investigation of this ncRNA will provide important insights and guide further research. In this review, authors provide a comprehensive overview of circHECTD1's expression and function across a spectrum of tumor and other diseases, elucidating its regulatory mechanisms and potential diagnostic and prognostic implications. This study, equip researchers with a swift grasp of the latest advancements in circHECTD1 research, emphasizing its pivotal role as a biomarker in disease diagnosis, treatment, and prognosis. This will contribute to filling the gaps in the literature on this topic. The review is well-written and comprehensive, but the following issues need to be addressed.
Minor typos are scattered throughout the text and should be corrected. Some examples include:
Page 8, line 115 (Error! Reference source not found.)
Page 11, lines 202, 219, 224, 226
Page 12, line 259
Page 23, line 706, etc.
Please check some references for errors such as "(Error! Reference source not found)."
On page 9, line 131, "MiRNAs" should be replaced with "miRNAs."
Some references might not have been used appropriately. For instance, on page 12, line 267.
On page 16, line 421, "3.3.2. silicosis" can be replaced with "3.3.2. Silicosis."
The font size of some titles does not appear to be consistent.
Authors are requested to provide the long forms of abbreviations when they are used for the first time.
An illustrative figure of circRNA function should be included under the title "2.2. The functions of circRNAs."
The authors did not refer to the figures they prepared in the text.
The title directly starts with "2.1. "1" missing?
The fourth title is not appropriate. "4. The roles of circHECTD1 in Diseases" should reflect the context-dependent nature of this circRNA. Otherwise, the authors addressed its role disease by disease in the manuscript.

Experimental design

The rationale behind the authors' selection of CircHECTD1 is unclear. It is important for the authors to provide a detailed explanation of why they chose to focus on this particular circRNA

Validity of the findings

The conclusion section admirably identifies unresolved questions, gaps, and future directions for research, highlighting the manuscript's potential to inspire further exploration and innovation in the field.

Reviewer 2 ·

Basic reporting

The pathological functions of circRNA come into focus in recent years. The manuscript by Yuan et al provides an updated review of a specific disease-relevant circRNA (circHECTD1) and discusses its potential clinical application. I have some concerns attached below.

1, To provide a more thorough analysis, it would be beneficial to assess the strengths and weaknesses of each study (at least for the representative ones) and provide personal insights and comments. This could include discussing limitations of experimental approaches or potential biases in the interpretation of results. Additionally, offering personal opinions and reflections on the findings could enhance readers' comprehension of the topic and prompt further discussion and research.
2, line 175 - 195 It is great to mention that certain circRNAs are able to encode functional proteins. Given this fact, the authors may consider providing a brief comment about the potential protein-coding capacity of circHECTD1 in the “perspectives” section. For example, using either cis- (see PMID: 28344080 & 28281539) or trans-acting (see PMID: 36330957) factors to engineer a controllable circHECTD1 translation tool that can produce pharmaceutical proteins in living organism.
3, Some sentences such as line41-42 are very vague and/or repetitive. Please reedit your manuscript to make it more concise.
4, Please carefully check and correct the grammar errors throughout the manuscript.

Experimental design

no comment

Validity of the findings

no comment

Additional comments

no comment

---

## Round 0.2 · Minor Revisions

Please address remaining issues indicated by the reviewers. Although one of the reviewers requested citing additional papers, you should do so only the case if you think that the recommended references are related to the subject of your article.

Reviewer 1 ·

Basic reporting

The authors have satisfactorily improved their paper in response to the comments. I would like to thank the authors for their great efforts. However, some minor issues need to be addressed before acceptance

1. Page 2, Line 63: circHECTED1 should be replaced with circHECTD1.

2. Please use a consistent format and capitalization for subtitles, e.g., "Gastric Carcinoma," "Hepatocellular carcinoma," "Ulcerative colitis," and "Acute Lung Injury."

3. Page 20, Line 775: The paragraph starting with "For example" and the sentences following it, including those starting with "Furthermore" and related to EMT, disrupt the context of the paragraph. The authors are requested to review and revise these sentences for clarity and coherence.

Experimental design

no comment

Validity of the findings

no comment

Additional comments

no comment

Reviewer 2 ·

Basic reporting

The manuscript has been well improved. However, before publication, the authors are suggested to cite the important papers that surveyed cis- or trans-acting regulators of circRNA translation. It will be useful for readers to provoke new research direction about circHECTD1.

Experimental design

no comment

Validity of the findings

no comment

Additional comments

no comment

---

## Round 0.3 · accepted · Accept

Since all remaining issues indicated by the reviewers were adequately addressed, the revised manuscript is acceptable now.